# *Tande nou gwonde*! (Hear us roar!)- Youth perspectives of maternal near-misses: Protocol for a photovoice study of young childbearing people's perspectives of maternal near-misses in northwest Haiti

Tonya MacDonald[1,2☉¤]*, Marie-Carmèle Charles[3☉], Olès Dorcely[4☉], Elizabeth K. Darling[1,2,5], Saara Greene[6], Sandra Moll[7], Carmen Logie[8,9,10], Lawrence Mbuagbaw[1,11,12,13,14,15☉]

1 Department of Health Research Methods, Evidence and Impact, McMaster University, Hamilton, ON, Canada, 2 McMaster Midwifery Research Centre, McMaster University, Hamilton, ON, Canada, 3 Montréal, QU, Canada, 4 Centre Médical Béraca, La Pointe, Haiti, 5 Department of Obstetrics and Gynecology, McMaster University, Hamilton, ON, Canada, 6 Faculty of Social Sciences, School of Social Work, McMaster University, Hamilton, ON, Canada, 7 School of Rehabilitation Science, McMaster University, Hamilton, ON, Canada, 8 Canada Research Chair in Global Health Equity and Social Justice with Marginalized Populations, Factor-Inwentash Faculty of Social Work, University of Toronto, Toronto, ON, Canada, 9 Centre for Gender & Sexual Health Equity, Vancouver, BC, Canada, 10 United Nations University Institute for Water, Environment & Health, Richmond Hill, ON, Canada, 11 Department of Anesthesia, McMaster University, Hamilton, ON, Canada, 12 Department of Pediatrics, McMaster University, Hamilton, ON, Canada, 13 Biostatistics Unit, Father Sean O'Sullivan Research Centre, St. Joseph's Healthcare, Hamilton, ON, Canada, 14 Centre for the Development of Best Practices in Health, Yaoundé Central Hospital, Yaoundé, Cameroon, 15 Division of Epidemiology and Biostatistics, Department of Global Health, Stellenbosch University, Cape Town, South Africa

☉ These authors contributed equally to this work.
¤ Current address: Department of Health Research Methods, Evidence & Impact, Health Sciences Centre, McMaster University, Hamilton ON, Canada
* macdota@mcmaster.ca

**Data Availability Statement:** Not applicable. There is no data available as this is a study protocol.

## Abstract

### Introduction

Globally, a shift is occurring to recognize the importance of young peoples' health and well-being, their unique health challenges, and the potential they hold as key drivers of change in their communities. In Haiti, one of the four leading causes of death for those 20–24 years old is pregnancy, childbirth, and the weeks after birth or at the end of a pregnancy. Important gaps remain in existing knowledge about youth perspectives of maternal health and well-being within their communities. Youth with lived experiences of maternal near-misses are well-positioned to contribute to the understanding of maternal health in their communities and their potential role in bringing about change.

### Objectives

To explore and understand youth perspectives of maternal near-miss experiences that occurred in a local healthcare facility or at home in rural Haiti.

**Funding:** The authors received no specific funding for this work.

**Competing interests:** The authors have declared that no competing interests exist.

**Abbreviations:** CBPR, Community-Based Participatory Research; CHL, Community Health Lead; CHW, Community Health Worker; CMB, Centre Médical Béraca; CRT, Community Research Team; HiREB, Hamilton Integrated Research Ethics Board; HIV, Human Immunodeficiency Virus; KM, Knowledge Mobilisation; KT, Knowledge Translation; LMIC, Low-to-Middle Income Country; LO, Local Obstetrician; LRC, Local Research Coordinator; MCC, Marie-Carmèle Charles; OD, Olès Dorcely; PHI, Personal Health Information; RA, Research Assistant; RL, Research Lead; TCPS, Tri-Council Policy Statement; TM, Tonya MacDonald; TBA, Traditional Birth Attendant; UN, United Nations; WHO, World Health Organization.

## Methods

We will conduct a qualitative, community-based participatory research study regarding maternal near-miss experiences to understand current challenges and identify solutions to improve community maternal health, specifically focused on youth maternal health. We will use Photovoice to seek an understanding of the lived experiences of youth maternal near-miss survivors. Participants will be from La Pointe, a Haitian community served by their local healthcare facility. We will undertake purposeful sampling to recruit approximately 20 female youth, aged 15–24 years. Data will be generated through photos, individual interviews and small group discussions (grouped by setting of near-miss experience). Data generation and analysis are expected to occur over a three-month period.

## Ethics and dissemination

Ethics approval will be sought from Centre Médical Béraca in La Pointe, Haiti, and from the Hamilton Integrated Research Ethics Board in Hamilton ON, Canada. We will involve community stakeholders, especially youth, in developing dissemination and knowledge mobilisation strategies. Our findings will be disseminated as an open access publication, be presented publicly, at conferences, and defended as part of a doctoral thesis.

## Introduction

For the first time, in 2016 the United Nations (UN) made a bold move to include *young people* alongside women and children in the UN Global Strategy of Women's, Children's and Adolescents' Health [1, 2]. This move signaled the recognition of young people having their own unique health challenges [1, 2] and highlighted the potential of youth as "key drivers of change in the post-2015 era" [2]. This shift signifies an important investment in young people (by UN definitions: adolescents age 10–24, and youth age 15–24) [3]) as full participants in society [1, 4]. Young people hold the opportunity to unleash the human potential of this "Sustainable Development Goals Generation" in order to transform our world [2]. Globally, a shift is occurring to recognize the importance of young peoples' health and well-being, and the need to systematically track progress of their health in order to improve it.

Haiti is a Low-Middle Income Country (LMIC) of the global south in which youth make up 21% of the population [5]. As a multi-burden country, Haiti has the highest maternal mortality ratio in the Western Hemisphere estimated at 488 maternal deaths per 100,000 live births [6]. According to Haitian national data from 2018, forty-two percent (42%) of births were attended by skilled birth personnel, thirty-nine percent (39%) of births were attended at health facilities, and about sixty-one percent (61%) of births took place in the home setting [7]. Haitian Traditional Birth Attendants (TBAs), called 'matwon' or 'fanm chay', support about fifty percent (50%) of the births in Haiti [7]. Currently, these TBAs are not supported by Haiti's public health ministry which exclusively advocates for "skilled birth attendance" in health facilities [8, 9]. Nonetheless, as valued, trusted, and knowledgeable birth attendants, 'matwon' support labouring individuals in the home setting, often in remotes areas that lack adequate medical and material resources [10–12]. Haiti has a high adolescent birth rate of 52 births per 1,000 women ages 15–19 [13]. Haitian adolescents and youth are faced with poor health outcomes [14]. One of the four leading causes of death for those 20–24 years old is pregnancy-related [15].

Haiti recognizes the need to collect maternal health data disaggregated by age and with inclusion of indicators specific to youth [16]. Haiti currently collects quantitative data as they relate, for example, to demand for modern methods of family planning/contraception, adolescent birth rate, births by age 18, antenatal care visits, skilled attendance at birth, obstetric complications and maternal mortality ratio [17]. Systematic tracking of *quantitative maternal youth indicators* has begun but an important gap remains in existing knowledge that links youth perspectives of maternal health and well-being within their communities [18]. There is limited information on the *qualitative factors* that influence maternal youth health and perinatal survival. How are gender constructs, such as gender roles, gender identity, and institutionalized gender at play amid these factors?

Maternal death among youth is an urgent problem in Haiti. There is a pressing need for understanding from young Haitian women and gender diverse childbearing people about who and why youth are dying during pregnancy, birth, and the weeks after. The maternal health experiences of young women and gender diverse childbearing people as seen through their eyes and heard through their voices is one step toward knowing more. This step holds the potential of addressing maternal health needs especially as they relate to youth empowerment within settings of constrained services and low resources.

An examination and exploration of youth maternal health outcomes and experiences warrants the use of various approaches and from different vantage points. One such vantage point comes from survivors of maternal near-misses. A maternal near-miss experience is when a person narrowly survives a grave obstetric event during pregnancy, birth or up until 42 days after birth [19]. In a comprehensive review Pacagnella and group examined the delays in obstetric care leading to maternal death to suggest that maternal near-misses, "events" in which severe pregnancy, childbirth or postpartum complications are averted (through timely/adequate care or by chance alone), are important in understanding maternal death itself [20]. These researchers believe this proxy group of near-misses represents a maternal group that often shares common characteristics as those who died "on the road of death" [20].

Maternal death can be accounted for quantitatively. However, only survivors of near-misses can offer a qualitative perspective of maternal near-death events. The lived experiences of these survivors, or "proxies", may be considered as a representation of both the voices of those who died en route for help and those who survived. Pacagnella et al. suggest that a broader range of information-gathering methods on maternal mortality may better address gaps in obstetric care and in turn positively affect maternal outcomes [20]. Youth with lived experiences of maternal near-misses are well-positioned to contribute to understanding of maternal mortality and improve maternal outcomes in their communities.

Research on maternal health in Haiti includes various aspects of maternal health related to women's experiences of pregnancy complications [21], their unmet health needs [22], factors influencing retention in Human Immunodeficiency Virus (HIV) care among adolescent girls and young women living with HIV in Haiti [23], and of Haitian women's barriers to and facilitators for cervical cancer prevention and control [24]. Some literature is specific to Haitian youth, and some studies have focused on addressing cultural orientation among both female and male adolescents [25], innovative health and sports interventions to reduce adolescent birth in Haiti [26], regarding risk and protective factors among HIV-affected youth [27] and of female and male youth perceptions of how the social context impacts their HIV risk, using Photovoice [28].

There is a paucity of research that identifies key measures focused on maternal health among youth and their lived experiences [28], and that includes multiple perspectives of young people who live in rural Haitian settings. Our study aims to support youth leadership and empower youth, to mobilize findings locally, and to inform maternal health policy and

practice. The goal of our study is to qualitatively explore young women and gender diverse childbearing people's knowledge of and perceptions of the contextual factors that influence their risk of maternal near-misses. By examining the social context (as impacted by constructs of gender), we hope to illuminate some of the underlying structural dimensions of risk and how social contexts influence young people's ability to negotiate aspects of their maternal health and well-being in low- and middle-income countries (LMICs) or low-resource countries. To the best of our knowledge, this is the first project that considers youth perspectives of maternal near-miss experiences in Haiti.

## Objectives

The objectives of this study are to understand and amplify the voices of Haitian youth who have experienced maternal near-misses:

### Primary objectives

- Engage Haitian youth in an arts-based Photovoice process of exploring and depicting their experiences of maternal near-misses in their community.

- Identify factors in the social environment that create barriers to youth maternal health and well-being.

- Explain how factors in the social environment increase the risk of maternal near-misses among youth.

- Characterize contextual factors that influence risk of maternal near-misses among youth.

- Support youth leadership and empower youth to share their findings regarding maternal health and well-being in their community.

- Mobilize the findings of youth to key stakeholders in order to advocate for improved maternal health and well-being among young women/gender diverse childbearing people.

### Secondary objectives

- Identify successes, and challenges regarding Photovoice methodology from the perspectives of project participants and researchers.

- Generate recommendations regarding Photovoice as an inclusive approach to optimize knowledge transfer and mobilisation.

### Research questions

Our research questions aim to explore the lived experiences and attitudes of a group of participants [29]- of young women and gender diverse childbearing people, living in La Pointe, Haiti, who have survived maternal near-misses. Our research questions:

1. What are the lived experiences of this group of young people?

2. What key challenges do they experience?

3. What possibilities do they see for change?

## Methods

### Study design

We will use Photovoice to explore the direct, lived experiences of youth maternal near-miss survivor participants from La Pointe, a community in northwest Haiti served by Centre Médical Béraca (CMB- a healthcare facility with community health outreach services [30] Our study design will be underpinned by philosophical foundations grounded in a transformative worldview and community-based participatory research (CBPR), through a lens of health equity, social justice, and sustainable community development [31]. This design centers the community and focuses on the need for social justice and human rights [31]. Our qualitative study is based on established relationships of a diverse team of Haitian and Canadian researchers and Haitian community members (**S1 Appendix**). The theoretical framing of our research aligns with these relational dynamics and is congruent with our CBPR approach, employing Photovoice, an arts-based methodology, *with youth* in their community.

### Photovoice methodology

Photovoice, as a participatory visual research methodology and method, is meant to foster social change and achieve empowerment of structurally vulnerable individuals and communities whose voices are not often heard [32–35]. This methodology was developed by Wang and Burris in the 1990s [36, 37]. Wang and Burris' Photovoice is underpinned by three main principles: empowerment, education, and participatory action [24, 28, 37, 38]. This methodology demands equitable and meaningful collaboration and is meant to bring citizens, researchers, and policy makers together to produce knowledge, for greater impact in communities [32, 37]. Photovoice has been used in healthcare, education, and in issues of policy and social justice, among others [39]. Health benefits relate to its being impactful as a participatory health promotion strategy and the potential to improve participants' lives by giving them voice and developing a critical awareness of their environment [39] which could in turn affect positive behaviour change in the community [28].

### Theoretical perspectives

Several theoretical perspectives are integral to CBPR and Photovoice methodology:

- Feminist theory that fosters empowerment and liberation [40];

- Paulo Freire's critical pedagogy that facilitates critical consciousness to stimulate meaningful change [41–43];

- Principles of documentary photography where visual representation is used for advocacy and social change [33, 37, 38].

Using a CBPR framework and Photovoice methodology [38, 43], we will take a dialectical pluralism perspective as a metaparadigm, driven by thoughtful integration of the different worldviews of involved study participants, community stakeholders and researchers [44]. We will emphasize a strengths-based and salutogenic orientation and interpretation of our work [45]. This perspective *recognizes the strengths* of youth- as survivors of near-miss experiences, and their willingness *to take action* in their community in order *to ameliorate health and well-being*. These worldviews will weave throughout the design to honour the values and amplify the voices of participants, stakeholders, and researchers, and focus on collaboration, empowerment, and positive change.

## Positionality statement

The first author (TM) as Research Lead (RL), approaches this study through a lens of equity, social justice, and sustainable community development. The RL understands the social justice underpinnings required of a community-based researcher, and the importance of "knowing who I am" [46: Hall, p.156] and being responsible for the actions taken within the research. As one of the insider-outsider co-researchers, the RL (she/her/elle) positions herself as a white, settler Canadian, middle-aged female who identifies as a woman, a mother, a teacher, a mid-wife, and a doctoral student. The RL is not a woman of colour, not Haitian, and not confronted with constant worry of maternal near-misses in her community. Previous work and research experiences in Haiti first led the RL to this community of interest that she continues to work with as a midwife educator and PhD student. As one of the co-researchers for this study, the RL acknowledges how she stands to gain academically and professionally from the completion of such a research project.

## Ethics

This research will have human participants, use photographs of their community, and will also use Personal Health Information (PHI) to identify potential participants. Our study will be guided by Tri-Council Policy Statement (TCPS): Ethical Conduct For Research Involving Humans [47]. Ethics approval has been received in writing from the Research Ethics Committee of Centre Médical Béraca, Haiti's Northwest Department of Health and the Hamilton Integrated Research Ethics Board (HiREB) [48] (project ID#: 14954) in Hamilton ON, Canada. Only participants who provide their written or marked, informed, and on-going consent to participate will be included in the study.

Our study will integrate WHO-recommended guidelines for the ethical and safe conduct of intervention research on domestic violence against women [49]. In addition, we will take a trauma- and violence-informed approach to this research, using a framework that directs particular ethical attention to participant safety [50]. We understand the potential for photographic images to be intrusive and to lead to unintended consequences. We will refer to guidelines that address ethical issues raised by the use of photovoice and ensure we exceed Wang and Redwood-Jones' minimum best practices of photovoice ethics [51].

Research participants will keep all their own digital photos taken during fieldwork. Photos they wish to share for the study (including for public display) will be transferred electronically, deidentified and saved with a unique study identifier or code number. To protect the privacy of those who might appear in photos (or their property), we will blur identifying aspects of photos. Transcripts of interviews and focus group discussions will be de-identified during transcription. Only designated person will be allowed to access the study key file and assign unique study identifiers or codes to participants. The study key file will be securely stored on MacDrive, a privately hosted, secure, cloud storage at McMaster University, Hamilton ON, Canada. Computer tablets for audio-visual files will be housed in a locked secure location in the CMB office of the Local Obstetrician (LO), a member of the community research team. At the end of the project, the community will have possession of the new knowledge (all the de-identified data) generated by the community. CMB will securely store these data. McMaster University will also have a copy of these data on the secure MacDrive. We will permanently delete all transcripts and translations five years after a community presentation or research paper publication. A digital copy of photos will be kept by the research team on the secure MacDrive when they are not displayed, and until five years after a community presentation or research paper publication, at which time they will be destroyed.

## Participants

**Sampling.** We will undertake purposeful sampling to recruit approximately 20 female youth (aged 15–24) with maternal near-miss experiences from La Pointe, Haiti, and surrounding communities served by CMB. Recruitment will include those who identify as young women or those who do not necessarily identify as "women" but who are biologically female (referred to as "gender diverse childbearing people") and have experienced a maternal near-miss. Some peer-driven snowball sampling will occur where recruited participants locate other potential participants [31].

Sampling will occur until we have obtained an in-depth understanding of youth's near-miss experiences among a sample of participants with a variety of experiences. Recruitment will involve two groups of participants: those with lived experience of a near-miss incident that occurred at CMB (Group A) or that occurred in the community "at home" (Group B). To identify cases of MNMs that occurred in a healthcare facility (Group A), we will first search hospital records for maternal cases that included conditions or complications such as severe postpartum hemorrhage, severe pre-eclampsia, eclampsia, sepsis or severe systemic infection, and ruptured uterus. MNM criteria would include clinical organ dysfunction (e.g., shock, uncontrollable fit, stroke), laboratory markers of organ dysfunction, or clinical management proxies (e.g., hysterectomy, blood transfusion, dialysis). To identify cases of MNMs that occurred at home (Group B), we will seek out participants who self-report having had a very grave obstetric incident during pregnancy, labour or childbirth, or the six weeks after the termination of their pregnancy, such that from their perspective, they narrowly escaped death due to some kind of intervention or help, outside of the healthcare system.

Initial recruitment is not anticipated to be a challenge but sustaining participation may pose some challenges. Difficulties of travel for participants within remote parts of the catchment will need to be accounted for. However, we will make every effort to go to participants for interviews and organize group discussions at convenient locations. To bound the study, we will limit participation to those with near-miss experiences that occurred anytime from March 2018 (date that coincides with the opening of CMB's new stand-alone maternity unit (see study [52]). Near-miss experiences will have occurred during pregnancy, in labour and birth, and/or the postpartum period up until about six weeks after giving birth or the end of a pregnancy. We will aim to have a similar number of participants with near-misses that took place in the healthcare facility (Group A) as those with near-misses at home (Group B).

**Recruitment.** Study information will be advertised on posters at key locations in the community and provided to members of the community for wider distribution and on in-hospital video streaming in waiting rooms. We will deliberate on how messaging on posters, videos and in phone scripts should be tailored as they relate to gender relations and institutionalized gender [53]. The LO will discuss sampling and recruitment with hospital staff and will examine hospital records to identify potential participants. Together with other members of the CRT, LO will confirm eligibility such that the sample of cases meets our operational definition of MNMs. Cell phone or face-to-face recruitment will be undertaken by our LRC. The LRC will communicate with former patients to ascertain their interest as Group A study participants. For recruitment of Group B participants, the LRC will use community networks to facilitate recruitment across CMB's catchment area and especially in the more remote areas of the catchment. This will occur by actively involving CMB Board members and their networks, local hospital staff, CMB's Community Health Lead, as well as community health workers (CHWs), traditional birth attendants (TBAs), healers, and especially leaders of youth clubs, groups, or teams. Key to this recruitment process will be visibility in places where youth are known to gather, and as an attempt to have maximal variation sampling (by different near-

miss maternal experiences such as timing or location of, gender identity, age, distance from healthcare facilities, and level of schooling).

To be included in the study, participants will need a cell phone with built-in camera or have access to one for temporary use. Given the transient nature of the population, we will exclude those planning to move outside Haiti in the three months from the anticipated start of the study. There are no other exclusion criteria except participants must be adequately fluent in Haitian Creole or French to participate in a Photovoice workshop, interviews, and group discussions (e.g., group analysis sessions).

**Sample size.**   Sample size will be informed by Wang and Burris's (1997) originally proposed sample size of 6–10 participants for the photovoice process [38, 54], with recognition that flexibility is important to be responsive to interest from the community and to ensure sufficient depth of understanding. We will consider the concept of saturation, but as stated by Braun and Clarke (2021), "attempting to predict the point of data saturation cannot be straightforwardly tied to the number of interviews (or focus groups) in which the theme is evident, as the meaning and indeed meaningfulness of any theme derives from the dataset, and the interpretative process" [55]. Our aim is for a sample size of approximately 20 participants (i.e., 10 per group), a sample size justified through Wang and Burris' photovoice process, but also supported by evidence from a scoping review of using photovoice as disability research method [54].

## Data generation and collection

Photovoice methodology will guide data generation and collection. This process will have participants take their own photos using a cell phone and later they will be asked to make choices about which photos most effectively represent their near-miss experiences. No instructions will be given regarding who, what, when, where, or how many photographs to take during their photo taking fieldwork [28], creating the opportunity for participants to be empowered to "find their own voice". However, during our Photovoice Workshop (prior to fieldwork) we will show examples from other photovoice studies, brainstorm ideas, and reassure participants of their autonomy in this phase of the research. After photography fieldwork, participants will be able to tell their photographic story of their individual experiences through individual semi-structured interviews. We will follow with a small group discussion (Groups A and B separately) to elicit the meaning of their selected photos. This collaborative discussion of meaning [37], including a visual display of each group's collective photos, will generate further data among the groups of youth. Data generation (and analysis) will be shaped by our research questions, and guiding questions for interviews and small group discussions. Textual data will also include observations and reflections of participant-CRT conversations and interactions, any participant field notes and CRT debriefings after interviews and discussions (data collection forms, interview guide, and participant pre-workshop and end-of-project questionnaires in **S1 File**). The Photovoice Workshop, interviews, and discussions will occur at a safe, private location.

**Procedures of photovoice.**   Over the course of approximately 8 weeks (optimally), participants will be involved in this Photovoice project. There is flexibility within the timeline for unforeseen issues that may emerge (timeline of photovoice procedures in **S1 Fig**). In weeks 1–2, participants will attend a Photovoice Workshop, learn about each other, and the Photovoice methodology, including training in the technicalities of taking photographs, ethical considerations and responsibilities of using a camera/camera phone, and picture ownership [56]. During weeks 2–3, participants will take photographs that correspond with the research questions, and the LRC will check-in with participants by phone to troubleshoot technical, ethical or practical challenges [24]. In weeks 4–5, participants will be individually interviewed. They will share their stories about their photographs and be asked to narrow down their photos to the ones that most

effectively represent their near-miss experiences. In weeks 6–7, participants will come together for a small group discussion of their collective photos and undertake a photo elicitation process. In week 8, participants and the CRT will finalize photographs and main themes for a curated Photovoice exhibition, for example. This process will be carried out for both Groups A and B; the process will only be undertaken with consenting participants at the start of the project, and with their ongoing individual and group consent throughout the study.

## Obtaining informed consent

In many jurisdictions (including the Province of Ontario, Canada), capacity for providing consent is not based on age, but on competency. In Haiti, according to national child protection legislation the age of consent is 15-years old [57]. A consent approach based on competency for those under age 16, adheres to national legislation and aligns with local expectations in this Haitian community. Thus, local researchers will confirm participant competency (i.e., individual's ability to understand what study participation means and ability to decide on study participation) and proceed with consent by capacity for participants who are age 15; they will not obtain parental consent. Since we anticipate issues of literacy for some of our intended population, we will use a script (in Haitian Creole) that outlines the risks and benefits of the study. We will read aloud a consent statement and provide a written copy of the study information and consent form. Participants will indicate consent either by signing or putting an "X" on consent forms. Participants will be reminded about voluntary participation and their rights to withdraw prior to or at any time during the project and confidentiality of the collected data.

During interviews and discussions, we will "check in" for continuing consent to participate. Participants will be reminded that they do not need to answer any questions that they don't want to, and they can stop the interview at any time or withdraw from the group discussion. By participating in interviews or group discussions, participants give permission for the research team to read their transcripts and use excerpts of transcripts. Photograph release consent forms will ensure that study participants understand what their photos will be used for, where their photographs will be published and how photos may be displayed in other knowledge mobilisation activities. With permission, participant-chosen pseudonyms (or names) will be used when showcasing photos and quotations in reports, exhibits, publications, etc.

We will safeguard community members who may appear in participants' photographs by providing a "media public release" form. Participants will use this form to obtain permission to photograph individuals in the community and/or their personal spaces/belongings. This will ensure these individuals understand the rationale behind photo collection and how they could be used. We will also offer to provide a copy of photos to individual community members, for their own keeping.

## Compensation

Each participant will be compensated for travel, their effort and time in data collection and their participation in workshop sessions, interviews, and discussions. Compensation will be: $10 USD/participant for each "activity": workshop, photo taking/interview and focus group discussion (i.e., a total maximum monetary compensation of $30 USD). Participants also be offered healthy snacks during workshop sessions, interviews and group discussions, a small gift of healthcare supplies, and a specific amount of data usage or calling minutes purchased in advance (for data and to access the CRT as needed). Upon project completion, participants will be invited for a project celebration where they will be presented with photographs they captured, and certificates of completion for study participants.

## Data analysis

Data analysis will be an iterative process (i.e., immersing in and returning to data often), throughout and after data generation and collection. We will remain open to the nature of the person's experience versus the location of the near-miss (i.e., home vs hospital). Analysis will be grounded in a visual and narrative approach. Interviews and group discussions will be recorded and transcribed verbatim in Haitian Creole by a (youth) community member with transcription and translation experience.

First, individual interviews will occur after photography fieldwork. Interviews will be semi-structured to ensure "space" for participants to describe their photographic stories. We will ask participants to give an overview of their captured images as an introduction. Then we will also ask questions that stimulate analysis and reflection such as, "How does this picture make you feel? or What message is the photo trying to convey?" [56: p. 879]. Participants will select the most meaningful photographs to them.

Next, small group discussions and photo elicitation will occur. These discussions will allow for collective interpretation of images by the group, shared interpretation of personal experiences and co-construction of meaning [37]. We will use the "SHOWED" guide to facilitate photo elicitation and to guide group analysis [37, 38, 43]. The guiding questions for the explication of each photo will include:

What do you **S**ee here?

What is really **H**appening here?

How does this relate to **O**ur lives?

**W**hy does this situation, concern, or strength exist?

How can we become **E**mpowered through our new understanding?

What can we **D**o about it? [37: p.5].

Finally, inductive narrative analysis of transcripts will take place. We will first analyze each participant's individual narrative in a way to preserve the particular histories of individuals [58, 59] and that align with their selected photographs. We will use an inductive approach to keep individual narratives intact. Guided by Riessman's process of narrative analysis, we will also look for similarities and differences between the participants' experiences [58]. We will also consider Eakin and Gladstone's "value-adding" approach to qualitative analysis [60], using "generative coding" to engage in a process of reflexivity and greater interaction with the narratives. To assist with data handling throughout analysis, we will upload narrative data into NVivo™ (version 14), a software for qualitative data analysis [61].

It should be noted that active involvement of youth participants in this study is both critical and central. This is particularly relevant during data generation, analysis and interpretation, and presentation of findings. Our approach to this project will be flexible and iterative to incorporate the perspectives, needs and suggestions of the youth participants and LRC throughout the project.

## Knowledge translation and mobilisation

The CRT will work in partnership to identify contextual knowledge gaps, to exchange knowledge regarding research questions/approach, project feasibility, and to develop knowledge translation (KT) and knowledge mobilisation (KM) strategies. The knowledge generated and translated from our study will be useful and of interest to several audiences. We intend to put

our gained knowledge into action, to mobilize our work on many fronts, widely and through engaged youth involved in the study.

We will keep youth participants who have been part of the research process, and co-created knowledge, central and pivotal to KT and KM development. We will plan to share findings with the wider community in a culturally appropriate and effective way, and to raise awareness of community maternal near-misses, especially among youth. We expect this to include curation of a photographic exhibition to the community at large but we will remain flexible to other youth-generated suggestions for KM activities. We will include some evaluation of the photovoice project and/or KT experience in terms of impact both on research participants and community members attending project KT events. Final study results will also be presented through targeted presentations and short videos.

## Impact

To date, no literature has been located that considers youth perspectives of maternal near-miss experiences. Neither has past research adequately involved youth voices to identify key measures focused on maternal health among youth, sensitive to this group and their lived experiences, and that includes multiple perspectives of young people who live in rural LMIC/Haitian settings. Further, the use of Photovoice to engage youth within the research process of understanding maternal near-misses in their community will be novel in this setting, among this population.

Our study will give participants an opportunity to engage in individual conversations, and to share their stories and photographs to a wider community about their near-miss maternal experiences. This study will engage, train, and empower local youth in maternal and community health promotion through this CBPR and using a Photovoice approach (with the potential for participants to increase their visual literacy of images and digital ethics around these). Our study will contribute new ways of knowing to the scientific community about the experiences of youth who have survived near-miss maternal experiences in a LMIC. This knowledge will also provide local data on maternal near-misses and help CMB and other community members understand how to provide better maternal health services for youth. This study will generate knowledge on the potential role of youth as an untapped resource to support maternal well-being in rural Haiti. And this will inform community members, especially youth, interested in social change toward improving maternal health and well-being.

## Supporting information

**S1 Appendix. Qualitative research design, strategy, and characteristics.**
(DOCX)

**S1 File. Data collection forms, interview guide, and participant pre-workshop and end-of-project questionnaires.**
(DOCX)

**S1 Fig. Timeline of photovoice procedures (in weeks).**
(TIF)

## Acknowledgments

We acknowledge and thank the new members of our Community Research Team, Rose Andrele Bien Aimé, Cherlande Joseph, Baldine Garçon and Dyna Bonne Anney Volcy, who have contributed their time, energy, and knowledge since the initial writing of this protocol.

We also extend our thanks to Martine Jean-Baptiste and Angie Menuau of the Foundation for Advancement of Haitian Midwives, for their support of our work.

## Author Contributions

**Conceptualization:** Tonya MacDonald, Marie-Carmèle Charles, Olès Dorcely, Elizabeth K. Darling, Saara Greene, Sandra Moll, Lawrence Mbuagbaw.

**Funding acquisition:** Tonya MacDonald, Lawrence Mbuagbaw.

**Methodology:** Tonya MacDonald, Marie-Carmèle Charles, Olès Dorcely, Elizabeth K. Darling, Saara Greene, Sandra Moll, Carmen Logie, Lawrence Mbuagbaw.

**Project administration:** Tonya MacDonald, Marie-Carmèle Charles, Olès Dorcely, Lawrence Mbuagbaw.

**Resources:** Tonya MacDonald, Marie-Carmèle Charles, Olès Dorcely, Elizabeth K. Darling, Saara Greene, Sandra Moll, Carmen Logie, Lawrence Mbuagbaw.

**Supervision:** Elizabeth K. Darling, Saara Greene, Sandra Moll, Carmen Logie, Lawrence Mbuagbaw.

**Writing – original draft:** Tonya MacDonald, Marie-Carmèle Charles, Olès Dorcely, Lawrence Mbuagbaw.

**Writing – review & editing:** Tonya MacDonald, Marie-Carmèle Charles, Olès Dorcely, Elizabeth K. Darling, Saara Greene, Sandra Moll, Carmen Logie, Lawrence Mbuagbaw.

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
