## [Decision Letter · Decision Letter 0]

8 Jan 2024

PONE-D-23-25467Tande nou gwonde! (Hear us roar!)- Youth perspectives of maternal near-misses: protocol for a photovoice study of young childbearing people’s perspectives of maternal near-misses in northwest HaitiPLOS ONE

Dear Dr. MacDonald,

Thank you for submitting your manuscript to PLOS ONE. After careful consideration, we feel that it has merit but does not fully meet PLOS ONE’s publication criteria as it currently stands. Therefore, we invite you to submit a revised version of the manuscript that addresses the points raised during the review process.

Please submit your revised manuscript by Feb 22 2024 11:59PM. If you will need more time than this to complete your revisions, please reply to this message or contact the journal office at plosone@plos.org. Please include the following items when submitting your revised manuscript:A rebuttal letter that responds to each point raised by the academic editor and reviewer(s). You should upload this letter as a separate file labeled 'Response to Reviewers'.A marked-up copy of your manuscript that highlights changes made to the original version. You should upload this as a separate file labeled 'Revised Manuscript with Track Changes'.An unmarked version of your revised paper without tracked changes. You should upload this as a separate file labeled 'Manuscript'.We look forward to receiving your revised manuscript.

Kind regards,

Joseph Adu, PhD, MSc., Mphil

Academic Editor

PLOS ONE

Journal Requirements:

3. We note that the original protocol that you have uploaded as a Supporting Information file contains an institutional logo. As this logo is likely copyrighted, we ask that you please remove it from this file and upload an updated version upon resubmission.

**Reviewer 1**

This is will be an interesting study if conducted, however, I have a few comments which if the authors address will help strengthen the protocol and the future study.

Introduction

In line 81, the authors should capitalize each word in the low-middle income country to read Low-Middle Income Country (LMIC)

In line 82, the reference [5] should come before the full stop. The same change should be made in 121 for reference [18].

Photovoice methodology

The description of the process of the photovoice, how many photographs will each participant take at a point?

Data collection

What is the operational definition for maternal near-misses in this particular study? What variable are authors looking at in defining this maternal near-misses? Will there be a review of maternal records from the health facilities? If yes. What about the participant that have not attended health facility but will be involved in this study?

Ethical issues

How will the photographs and other data collected by the participants be handled, stored and managed to ensure the confidentiality and privacy of the participants themselves and those that might appear in the photographs? And what will happen to those photographs taken after the study?

How will the consenting of the participants who could be under 18 years old be taken? Or the authors do not anticipate that? This is a case where the study is among young women

Analysis

How will the analysis of the narrative of transcripts be done? Will there be the use of any qualitative analysis software? If there is one, which one will be used?

**Reviewer 2**

This is a very well written research protocol on an important and understudied topic. The authors make a clear justification for why the collection of qualitative data about maternal near-misses among youth in Haiti is warranted (i.e.: survivors of maternal near-misses act as a ‘proxy’ group). Definitions of youth/adolescence are well described as is the explanation for including gender diverse childbearing people in recruitment. Excellent justification for the use of Photovoice methodology and CBPR framework. Good inclusion of a positionality statement. The authors provided a very detailed and thoughtful explanation about the empowering and emancipatory potential of this research study.

Below are some minor comments/suggestions:

Introduction:

Lines 85-86: The sentence: “One of the four leading causes of death for those 20-24 years old is pregnancy, childbirth and the weeks after birth or at the end of a pregnancy” is awkward. Could it be re-worded to say: “One of the four leading causes of death for those 20-24 years old is pregnancy, childbirth and the weeks after birth”.

Sampling:

Lines 233-234: Is there any information that can be included as to whether more births in Haiti occur in healthcare facilities or in the community / at home?

Recruitment:

Access to a cell phone should be listed as an inclusion criterion to participate. Is it assumed that the majority of youth (of all socioeconomic statuses) in Haiti have access to cell phones with adequate storage for photos and access to data plans and chargers for their phones?

Compensation:

Line 319: Nice explanation. Is the total maximum monetary compensation USD$30? (workshop, photo taking/interview and focus group discussion)? Please provide some context about the small gift of cellphone minutes.

Reviewers' comments:

Reviewer's Responses to Questions

**Comments to the Author**

1. Does the manuscript provide a valid rationale for the proposed study, with clearly identified and justified research questions?

Reviewer #1: Yes

Reviewer #2: Yes

2. Is the protocol technically sound and planned in a manner that will lead to a meaningful outcome and allow testing the stated hypotheses?

Reviewer #1: Yes

Reviewer #2: Yes

3. Is the methodology feasible and described in sufficient detail to allow the work to be replicable?

Reviewer #1: Yes

Reviewer #2: Yes

4. Have the authors described where all data underlying the findings will be made available when the study is complete?

Reviewer #1: Yes

Reviewer #2: Yes

5. Is the manuscript presented in an intelligible fashion and written in standard English?

Reviewer #1: Yes

Reviewer #2: Yes

6. Review Comments to the Author

You may also provide optional suggestions and comments to authors that they might find helpful in planning their study.

Reviewer #1: This is will be an interesting study if conducted, However, I have a few comments which if the authors address will help strengthen the protocol and the future study.

Introduction

In line 81, the authors should capitalize each word in the low-middle income country to read Low-Middle Income Country (LMIC)

In line 82, the reference [5] should come before the full stop. The same change should be made in 121 for reference [18].

Photovoice methodology

The description of the process of the photovoice, how many photographs will each participant take at a point?

Data collection

What is the operational definition for maternal near-misses in this particular study? What variable are authors looking at in defining this maternal near-misses? Will there be a review of maternal records from the health facilities? If yes. What about the participant that have not attended health facility but will be involved in this study?

Ethical issues

How will the photographs and other data collected by the participants be handled, stored and managed to ensure the confidentiality and privacy of the participants themselves and those that might appear in the photographs? And what will happen to those photographs taken after the study?

How will the consenting of the participants who could be under 18 years old be taken? Or the authors do not anticipate that? This is a case where the study is among young women

Analysis

How will the analysis of the narrative of transcripts be done? Will there be the use of any qualitative analysis software? If there is one, which one will be used?

Reviewer #2: This is a very well written research protocol on an important and understudied topic. The authors make a clear justification for why the collection of qualitative data about maternal near-misses among youth in Haiti is warranted (i.e.: survivors of maternal near-misses act as a ‘proxy’ group). Definitions of youth/adolescence are well described as is the explanation for including gender diverse childbearing people in recruitment. Excellent justification for the use of Photovoice methodology and CBPR framework. Good inclusion of a positionality statement. The authors provided a very detailed and thoughtful explanation about the empowering and emancipatory potential of this research study.

Below are some minor comments/suggestions:

Introduction:

Lines 85-86: The sentence: “One of the four leading causes of death for those 20-24 years old is pregnancy, childbirth and the weeks after birth or at the end of a pregnancy” is awkward. Could it be re-worded to say: “One of the four leading causes of death for those 20-24 years old is pregnancy, childbirth and the weeks after birth”.

Sampling:

Lines 233-234: Is there any information that can be included as to whether more births in Haiti occur in healthcare facilities or in the community / at home?

Recruitment:

Access to a cell phone should be listed as an inclusion criterion to participate. Is it assumed that the majority of youth (of all socioeconomic statuses) in Haiti have access to cell phones with adequate storage for photos and access to data plans and chargers for their phones?

Compensation:

Line 319: Nice explanation. Is the total maximum monetary compensation USD$30? (workshop, photo taking/interview and focus group discussion)? Please provide some context about the small gift of cellphone minutes.

7. PLOS authors have the option to publish the peer review history of their article (what does this mean?). If published, this will include your full peer review and any attached files.

Reviewer #1: No

Reviewer #2: No

---

## [Author Response · Author response to Decision Letter 0]

22 Mar 2024

We have included our detailed responses to the reviewers in our submission file called 'Response to reviewers.docx'.

---

## [Decision Letter · Decision Letter 1]

4 Apr 2024

PONE-D-23-25467R1Tande nou gwonde! (Hear us roar!)- Youth perspectives of maternal near-misses: protocol for a photovoice study of young childbearing people’s perspectives of maternal near-misses in northwest HaitiPLOS ONE

Dear Dr. MacDonald,

Thank you for submitting your manuscript to PLOS ONE. After careful consideration, we feel that it has merit but does not fully meet PLOS ONE’s publication criteria as it currently stands. Therefore, we invite you to submit a revised version of the manuscript that addresses the points raised during the review process.

We look forward to receiving your revised manuscript.

Kind regards,

Joseph Adu, PhD, MSc., Mphil

Academic Editor

PLOS ONE

Journal Requirements:

Reviewers' comments:

Reviewer's Responses to Questions

**Comments to the Author**

1. Does the manuscript provide a valid rationale for the proposed study, with clearly identified and justified research questions?

Reviewer #1: Yes

2. Is the protocol technically sound and planned in a manner that will lead to a meaningful outcome and allow testing the stated hypotheses?

Reviewer #1: Yes

3. Is the methodology feasible and described in sufficient detail to allow the work to be replicable?

Reviewer #1: Yes

4. Have the authors described where all data underlying the findings will be made available when the study is complete?

Reviewer #1: Yes

5. Is the manuscript presented in an intelligible fashion and written in standard English?

Reviewer #1: Yes

6. Review Comments to the Author

You may also provide optional suggestions and comments to authors that they might find helpful in planning their study.

Reviewer #1: I commend you on your efforts to address all the previous comments, however, I still have a few clarifications and suggestions. I pray you can accept this in good faith and address them

Line 98: Are the Haitian Traditional Birth Attendants (TBAs) an officially recognised part of the Haitian health system? Are they regulated? If yes, it should be indicated, or otherwise, a statement of clarification should be indicated of their clandestine and unauthorized nature

Line 296: The full meaning of LO should be provided when being used for the first time before the subsequent use of the abbreviation

Ethics

I am still not satisfied by the justification given on the consent/parental ascent for participants that are below 16 years

You mention that "In Haiti the capacity to provide consent is likewise not based on age but on the

person’s competency" It will be very helpful if the authors can provide any reference to this statement

7. PLOS authors have the option to publish the peer review history of their article (what does this mean?). If published, this will include your full peer review and any attached files.

Reviewer #1: **Yes: **Robert Kokou Dowou

---

## [Author Response · Author response to Decision Letter 1]

12 Apr 2024

We have responded to Reviewer #1 comments in the uploaded document "Response to Reviewers".

---

## [Editor Report · Decision Letter 2]

22 Apr 2024

Tande nou gwonde! (Hear us roar!)- Youth perspectives of maternal near-misses: protocol for a photovoice study of young childbearing people’s perspectives of maternal near-misses in northwest Haiti

PONE-D-23-25467R2

Dear Dr. Tonya Ann MacDonald

We’re pleased to inform you that your manuscript has been judged scientifically suitable for publication and will be formally accepted for publication once it meets all outstanding technical requirements.

Kind regards,

Joseph Adu, PhD

Academic Editor

PLOS ONE
---

## [Editor Report · Acceptance letter]

30 Apr 2024

PONE-D-23-25467R2 

PLOS ONE

Dear Dr. MacDonald, 

I'm pleased to inform you that your manuscript has been deemed suitable for publication in PLOS ONE. Congratulations! Your manuscript is now being handed over to our production team.

Kind regards, 

on behalf of

Dr Joseph Adu 

Academic Editor

PLOS ONE